# MAIT Cells: Partners or Enemies in Cancer Immunotherapy?

**DOI:** 10.3390/cancers13071502

**Published:** 2021-03-25

**Authors:** Dasha T. Cogswell, Laurent Gapin, Heather M. Tobin, Martin D. McCarter, Richard P. Tobin

**Affiliations:** 1Department of Surgery, Division of Surgical Oncology, University of Colorado Anschutz Medical Campus, Aurora, CO 80045, USA; Martin.Mccarter@cuanschutz.edu; 2Department of Immunology and Microbiology, University of Colorado Anschutz Medical Campus, Aurora, CO 80045, USA; Laurent.Gapin@cuanschutz.edu; 3Department of Medicine, University of Colorado Anschutz Medical Campus, Aurora, CO 80045, USA; Heather.Tobin@cuanschutz.edu

**Keywords:** MAIT cell, immunotherapy, unconventional T cell, cancer, MR1

## Abstract

**Simple Summary:**

Unconventional T cells have recently come under intense scrutiny because of their innate-like effector functions and unique antigen specificity, suggesting their potential importance in antitumor immunity. MAIT cells, one such population of unconventional T cell, have been shown to significantly influence bacterial infections, parasitic and fungal infections, viral infections, autoimmune and other inflammatory diseases, and, as discussed thoroughly in this review, various cancers. This review aims to merge accumulating evidence, tease apart the complexities of MAIT cell biology in different malignancies, and discuss how these may impact clinical outcomes. While it is clear that MAIT cells can impact the tumor microenvironment, the nature of these interactions varies depending on the type of cancer, subset of MAIT cell, patient demographic, microbiome composition, and the type of therapy administered. This review examines the impact of these variables on MAIT cells and discusses outstanding questions within the field.

**Abstract:**

A recent boom in mucosal-associated invariant T (MAIT) cell research has identified relationships between MAIT cell abundance, function, and clinical outcomes in various malignancies. As they express a variety of immune checkpoint receptors and ligands, and possess strong cytotoxic functions, MAIT cells are an attractive new subject in the field of tumor immunology. MAIT cells are a class of innate-like T cells that express a semi-invariant T cell antigen receptor (TCR) that recognizes microbially derived non-peptide antigens presented by the non-polymorphic MHC class-1 like molecule, MR1. In this review, we outline the current (and often contradictory) evidence exploring MAIT cell biology and how MAIT cells impact clinical outcomes in different human cancers, as well as what role they may have in cancer immunotherapy.

## 1. Introduction

Mucosal-associated invariant T (MAIT) cells are a class of innate-like T cells that exist in a pre-primed memory state and express a semi-invariant T cell antigen receptor (TCR) that recognizes microbially derived non-peptide antigens presented by the non-polymorphic MHC class-1 like molecule, MR1 [1]. With MAIT cells possessing many functions often associated with successful antitumor immune responses, they represent an attractive population to explore with respect to their potential roles in antitumor immunity. However, MAIT cells are a controversial topic in the field of tumor immunology with studies showing conflicting results in regard to whether they contribute to tumor growth, tumor regression, or play a neutral role in human cancers.

Early investigations of MAIT cells in cancer suggested they may represent a potential positive prognostic indicator. Indeed, screening of ~18,000 human tumors across 39 malignancies found a significant association of the KLRB1 gene (encoding CD161, a marker of MAIT cells) and favorable prognosis [2]. However, CD161 is also expressed on other immune cells, including conventional CD8+ T cells and natural killer (NK) cells [3], rendering the association with MAIT cells uncertain. Since then, other MAIT cell signatures have been used to examine the potential role, if any, that MAIT cells might play in tumor immunology. The results suggest that MAIT cells may play a significant role in tumor immunity, however considering recent discoveries as well as the other studies reviewed below, that role appears to be complex and often contradictory. In this review, we outline MAIT cell interactions with different types of cancers, focusing on studies in human cancer patients, and attempt to understand the future therapeutic potential of MAIT cells.

## 2. MAIT Cell Basics

### 2.1. Identification, Prevalence, and Development

Human MAIT cells express the TCR gene segment Vα7.2 (also known as TRAV1-2) joined to Jα33 (also known as TRAJ33) and a restricted repertoire of TCRβ chains [1,4]. Initial studies relied on antibodies specific for the Vα7.2-Jα33 TCR and the CD3 co-receptor, with a combination of various surface proteins, such as CD161, CD26, CD218 (IL-18 receptor), and MDR1 for the identification of MAIT cells [5,6,7,8,9,10,11,12,13,14,15] (Figure 1). However, MAIT cells are currently identified using MR1 tetramers loaded with the bacterial ligand 5-(2-oxopropylideneamino)-6-D-ribitylaminouracil (5-OP-RU) (MR1-tet) [16,17]. The differences in which markers were used to identify MAIT cells complicates the interpretation of some of these results. Most studies used a combination of Vα7.2-Jα33, CD3, CD161, with only a handful using the MR1 tetramer for a definitive identification [18,19,20,21,22,23,24]. The introduction of the MR1 tetramer has allowed for improved accuracy in identifying MAIT cells, as there are cells that express the Vα7.2 TCR, as well as CD161, that do not recognize MR1 antigens and are not MAIT cells [25].

MAIT cells are divided into subsets based upon the expression of the CD4 and CD8 co-receptors. The most common subset is CD4−CD8+ MAIT cells, constituting approximately 80% of total circulating MAIT cells in healthy adults [26]. The CD4−CD8− subset constitutes about 15% of the total MAIT cells, and CD4+CD8− and CD4+CD8+ subsets represent less than 5% of total circulating MAIT cells [26]. It is evident that results vary depending on which subset of MAIT cell is analyzed. In the studies discussed below, only five evaluated the frequency and/or function of the CD4+ MAIT subset [9,12,20,21,27] (Table 1). Studies that only report the activity of CD8+ MAIT cells may be overlooking this important subset that could enhance our understanding of MAIT cell biology in cancer patients. The functional differences and post-stimulatory effects of each subset have yet to be fully elucidated, but it is likely that they have varying roles to play in tumor immunity.

MAIT cells can also express certain transcription factors making them distinct from other conventional CD4+ and CD8+ T cells, such as promyelocytic leukemia zinc finger protein (PLZF) (encoded by ZBTB16), eomesodermin (encoded by EOMES), RAR-related orphan receptor γt (RORγt) (encoded by RORC) and T-bet (encoded by TBX21) (Figure 1). In mice, different functional subsets of MAIT cells have been found based on distinct expression of T-bet and RORγt, respectively. T-bet-expressing MAIT1 cells predominantly produce IFN-γ upon stimulation, while RORγt-expressing MAIT17 cells are biased towards IL-17 production [29]. Similar patterns have been found in humans [30,31], however, confirmation of such functionally specialized subsets requires further analysis. Despite their expression of RORγt, human MAIT cells mostly produce IFN-γ when stimulated [15,32,33], and further studies are required to precisely determine what circumstances lead to IL-17 production.

MAIT cells specifically recognize metabolic intermediates generated by microbial riboflavin biosynthesis, which play an essential role in MAIT cell development, activation, maintenance, and proliferation. MAIT cells are selected in the thymus on double positive thymocytes in an MR1-dependent fashion, and then proliferate in the periphery when exposed to microbial antigens, often from host microbiota [34]. In humans, MAIT cells were first described in the intestinal mucosa and liver [35]; however, they can be found in many other organs, constituting up to 10% of circulating CD8+ T lymphocytes and 5% of total blood T cells in healthy adults [17,36]. Due to their significant enrichment in mucosal tissues, it was initially believed that MAIT cells would primarily be involved with mucosal protection against viral or bacterial pathogens, however as our understanding of MAIT cells has evolved, it is clear that they are an abundant cell population possessing many different functions, some of which have undoubtedly yet to be discovered.

In addition to changes within anatomical site, age-related decreases are observed, with MAIT cells expanding in the periphery until approximately age 25, and then steadily decreasing over time until they comprise less than 1% of total T cells at age 70 [37,38,39]. In human fetal tissues, MAIT cells are enriched in the small intestine, liver, and lung, and proliferate rapidly when exposed to bacterial antigens. It is in the fetal mucosa that MAIT cells acquire an innate response to bacterial stimuli before they are exposed to commensal microbiota or environmental microorganisms, a mechanism in place to potentially help newborn immune systems distinguish between commensal and pathogenic microbes [40]. The mechanism behind this is still unclear, as it has not been shown whether fetal MAIT cell development happens through an endogenous MR1 ligand or from commensal bacterial metabolites released from maternal sources [40]. After birth, a critical period exists in which riboflavin-producing microbial exposure to MAIT cells is essential for proper development, as studies have shown that microbe colonization in older germ-free mice does not promote healthy MAIT cell development [41]. Bacteria capable of riboflavin biosynthesis are necessary for MAIT cell development, as mice lacking these bacteria also lack MAIT cells [34,42]. These observations highlight an important role for the microbiome and microbiota-derived products in the development of a healthy and immunocompetent organism.

### 2.2. Antigen-Specificity and Activation

There are several different microbially derived antigens that, when bound to MR1, are capable of either enhancing or inhibiting MAIT cell activation [43,44,45,46]. Strong activators include the compound 5-OP-RU as the most potent activator (Figure 2A), while 6-formyl pterin (6-FP), a formic acid metabolite, is often cited as an inhibitor of MAIT cell activation [16,43,45,47]. In addition to MR1-dependent activation, MAIT cells can be activated in an MR1-independent manner from virally infected cells or other immune cells via the cytokines IL-12, IL-18, and IL-15, with IL-15 activation dependent on the presence of IL-18 [48,49] (Figure 2B).

To date, no endogenous (host-derived) ligands have been identified that activate MAIT cells in an MR1-dependent or independent manner, however the thymus of germ-free mice is still able to select MAIT cells during development (even though the process is impaired) [32], hinting at the existence of such ligands. Additionally, supporting this theory is the presence of low-level endogenous cell surface expression of MR1 in humans and human cell lines [45,50]. As MR1 is stored in the endoplasmic reticulum until a ligand binds prompting its migration to the surface of the cell [51], this observation questions the source and identity of this ligand.

In humans, 80–90% of MAIT cells are skewed towards Th1-like functions, rapidly secreting IFN-γ and TNF-α upon activation [33,52]. These T-cell derived cytokines are often been associated with a better prognosis in most cancers [53]. In contrast, a relatively small population (~3%) of MAIT cells produce Th17/Th2-like cytokines including IL-17, which can have both pro-tumor and anti-tumor effects [52,54,55,56,57,58,59]. There is evidence indicating that IL-17 can promote tumor proliferation [60,61], angiogenesis [62,63,64], metastasis [65], and invasion [65], as well as enhance immune evasion by tumor cells [59]. Furthermore, a high Th1 to Th17 ratio has also been associated with a better prognosis for patients with colorectal cancer [66]. Conversely, evidence also shows IL-17 promotes migration of NK cells [67], neutrophils [68], and cytotoxic lymphocytes [69] to the tumor tissue and inhibits tumor cell invasion [70]. It is important to note that quantities of IL-17 are correlated with poor prognosis while high Th17 cell frequencies are correlated with improved prognosis, possibly explaining these discrepancies and highlighting the need to differentiate between Th17 and IL-17 producing cells [71].

### 2.3. TCR-Dependent Activation vs. TCR-Independent Activation

The manner in which MAIT cells are activated results in distinct transcriptional programs. TCR-dependent activation results in increased expression of RORγt [30,31], which is consistent with the mouse data showing an RORγt-based IL-17 bias [15], but there are conflicting reports showing both TCR-dependent increases and decreases in PLZF and T-bet [30,31], limiting the translatability from mice (Figure 2A). In TCR-independent activation, elevated expression of T-bet has been shown, again consistent with the mouse data showing a T-bet-associated IFN-γ bias (Figure 2B). PLZF levels in TCR-independent activation, again with conflicting reports, are unclear [30,31] (Figure 2B). These changes in transcriptional profile result in distinct activation-dependent cytokine profiles. When MR1 is engaged, MAIT cells predominantly release TNF-α and IL-17a, while under IL-12/IL-18 activation MAIT cells mostly release IFN-γ (Figure 2) [30,31]. It is unknown whether MAIT cells can shift between these two phenotypes or what may influence such changes.

Both TCR-dependent and TCR-independent modes of activation can trigger degranulation (as measured by degranulation marker CD107a) [30], releasing granzyme (GZM) A, GZMB, GZMK, GZMH, GZMM, and perforin on target cells [30,72] (Figure 2). Notably, granzyme A and K were downregulated in MAIT cells stimulated with *E. coli* or 5-A-RU (5-amino-6-D-ribitylaminouracil, the precursor to 5-OP-RU) (Figure 2A) but upregulated in response to IL-12+IL-18 treatment, and both activation modes upregulated secretion of GZMB, GZMH, and perforin [30] (Figure 2B). Further, TCR-independent activation can also release GZMM (Figure 2B). GZMK is of particular interest in MAIT cell research because it has the potential to induce apoptosis in a caspase-independent manner [73]. As tumor cells often become resistant to typical apoptotic pathways [74], this observation makes MAIT cells an intriguing potential ally for cancer immunotherapy.

A unique property of TCR-dependent activation is the capacity to secrete molecules to mediate tissue repair [75], underscoring the importance of the microbiota in promoting tissue homeostasis (Figure 2A). Damage to the mucosal barrier, and subsequent exposure to microbes, can recruit MAIT cells to the damaged site and drive secretion of tissue repair molecules like VEGF, TGFβ, HIF1α, FURIN, PTGES2, and PDGFB [75]. In mice, TCR-activated MAIT cells were able to accelerate wound closure, which was further strengthened by the topical application of 5-OP-RU [41]. If malignancies are viewed as a “wound” in the body, MAIT cells may help with immune surveillance in part through this mechanism.

## 3. MAIT Cells and Cancer

### 3.1. Mucosal-Associated Cancers

As microbial ecosystems and MAIT cells are so prevalent in mucosal tissues, researchers have taken a great interest in the role of MAIT cells in these types of cancers. Several studies of colorectal cancer (CRC) patients have found increased frequencies of MAIT cells within malignant tissues compared to healthy adjacent tissue [6,7,8,9,10], with corresponding decreases in the circulation [9,10], suggesting infiltration and accumulation of MAIT cells into the disease site (Table 1). However, other studies looking at CRC metastases found decreased frequencies of MAIT cells in liver metastases of CRC patients compared to healthy adjacent liver tissue [23] (Table 1). This could indicate that there are differences in MAIT cell infiltration of primary tumor versus metastatic sites, or that perhaps the primary tumor in CRC houses more MAIT cell-activating bacterial products than liver tissue. It is also possible that MAIT cells in the metastases are exhausted or rendered ineffective. In any case, more research on this topic is required.

Once activated, tumor-infiltrating MAIT cells display decreased IFN-γ and TNF-α production, with corresponding increases in IL-17 production [7,9,23] (Table 1) (Figure 3). IL-17 has a complicated function within the tumor microenvironment (TME), as it is known to aid in wound healing as well as contribute to tumor angiogenesis [59,76]. Another cytokine recently discovered to be secreted by chronically activated MAIT cells is IL-13 [77] (Figure 3), which has properties similar to IL-17, in that IL-13 can also promote tumor growth and is an important player in tissue repair. In mucosal cancers, tumor growth can cause intestinal barrier damage, which could be part of the reason that IL-17 and IL-13 are initially released and then contribute to the tumor-promoting qualities of the TME.

In mice, MAIT cells display a bias toward IL-17 production in mucosal tissues [29]. If this is also true in humans, it could explain the variability seen in mucosal malignancies versus other types of cancers. In addition, bronchoalveolar MAIT cells from children with severe pneumonia produced more IL-17 than either less severe cases or from circulating MAIT cells [78]. It is currently not clear whether the release of IL-17 is a direct result of chronic activation, severity of illness, location in the body, a part of the normal sequence of MAIT cell maturation, or a combination of these factors.

A recent CRC study noted that the tumor-residing MAIT cell exhausted phenotype is characterized by TCR-induced CD39 expression [24] (Figure 3). This is consistent with other work showing that chronic TCR stimulation on intra-tumoral MAIT cells leads to decreased responsiveness and upregulation of PD-1 and exhaustion markers CD39 and *CXCL13*, [79,80] (Figure 3), demonstrating tumor-infiltrating MAIT cell dysfunction associated with the TME.

The mechanisms resulting in MAIT cell activation within the TME are still not fully understood. It has been suggested that because CRC tumors are in close contact to the microbe-inhabited layers of the intestinal tract, the damage to the intestinal barrier caused by tumor growth could initiate a TCR-dependent, riboflavin-induced response in tissue-resident or infiltrating MAIT cells [8,77]. If this is the case, it could explain why tumor-infiltrating MAIT cells are observed in a chronic state of activation and exhaustion, perhaps contributing to the detrimental TME-induced changes in MAIT cell functionality. There is less evidence for activation via a tumor-derived antigen, but this possibility cannot be excluded, and these antigens, if discovered, would provide ideal targets for immunotherapies. However, IL-12 and IL-18 are present in colon and CRC tissue [81], so TCR-independent activation is possible as well, if not a combination of all of the above. Future studies investigating pre-cancerous colon polyps and patient progression/survival data may shed some light on the roles MAIT cells play in CRC progression.

Studies investigating esophageal cancers also observed decreased MAIT cell frequencies in the blood, with increased frequencies within esophageal tumors, as well as reduced expression of IFN-γ and TNF-α and no changes in IL-17 [14] (Table 1). Interestingly, this study found the risk of death was inversely correlated with the frequency of MAIT cells within the TME [14], indicating that, in this case, MAIT cell infiltration may assist in immune surveillance. Links have also been made between microbiome composition, antibiotic use, and the severity of esophageal cancers, suggesting that bacteria inhabiting the esophagus could impact the MAIT cell response to esophageal cancer cells [82]. Analyses of data from The Cancer Genome Atlas (TCGA) associated high MAIT cell gene signatures with improved overall survival and progression free survival in esophageal cancer patients [80]. However, this same analysis found worse overall survival in patients diagnosed with CRC and lung squamous cell carcinoma with increased MAIT cell signatures [80].

There is conflicting evidence regarding cervical cancer (CC), as some report decreased MAIT cells in the blood [13] and others report increased MAIT cells in the blood of CC patients [27] compared to healthy individuals (Table 1). In addition to more total MAIT cells, the latter study also found increased CD8+ MAIT cells, CD4+ MAIT cells, and highly activated CD38+CD8+ MAIT cells in CC patients [27] (Figure 1, Table 1). Additionally, a positive correlation was found between total MAIT cells and myeloid-derived suppressor cells (MDSCs) [27], a class of cells that have been shown to suppress cytotoxic abilities [83] and proliferation of T cells [84], positively correlate with tumor burden [84], promote disease progression [83], and decrease the overall survival of melanoma patients [83]. It has been hypothesized that MAIT cells, via secretion of IL-17 and/or other pro-inflammatory cytokines, could attract MDSCs and cause accumulation in the tumor site [85]. However, less double negative (DN) MAIT cells and PD-1+DN MAIT cells were found in these patients, and there was a positive association between high circulating PD-1+ DN MAIT cells and progression free survival (PFS). These findings are a strong reminder that there are not yet clear tumor-promoting or tumor-killing distinctions, and MAIT cell effects on tumor progression and survival change based on the subset or tissue analyzed.

### 3.2. Hepatic Cancers

Hepatocellular carcinomas (HCC) generally exhibit decreased MAIT cell frequencies compared to healthy adjacent liver tissue [11,19,80], with corresponding decreases in peripheral circulation [19,80], contrary to the mucosal cancers discussed above (Table 1). A possible mechanism accounting for this observation could be via the migratory receptors CCR9, CCR6, and CXCR6 which are significantly downregulated on MAIT cells in the circulation and TME in patients with HCC (Figure 1). Although MAIT cells in HCC patients already express lower levels of these receptors in peripheral blood compared to healthy donors, they are greatly reduced within the liver tissue and even further reduced near malignant liver tissue [19]. High levels of these receptors on T cells correlate with prolonged overall survival in melanoma patients [86], while mice lacking these receptors show enhanced tumor progression [86]. This could mean that the low levels observed in HCC may be representative of defective MAIT cell trafficking or a response to immunosuppressive signals from the TME. MAIT cells isolated from HCC tumor tissue support the latter hypothesis, expressing higher levels of inhibitory molecules such as PD-1, CTLA-4, TIM-3, and lower levels of IFN-γ, IL-17, granzyme B, and perforin [19] (Table 1, Figure 3). These cells also express more pro-tumor IL-8, which has been shown to recruit MDSCs to the TME [83] (Figure 3). Given the negative effects that MDSCs have on tumor progression, this observation presents an intriguing potential mechanism for the detrimental effects of MAIT cells in HCC.

When cultured in vitro with HCC cells, MAIT cells isolated from HCC tumors promote proliferation, invasion, and inhibit apoptosis in cancerous cells, a response contrary to results seen from a culture of MAIT cells derived from healthy donors [19]. These results show that the TME can induce markers of functional exhaustion, promote tumor angiogenesis, and cause defects in tumor killing capacity. Moreover, transcriptomic analyses revealed expression differences in tumor-infiltrating MAIT cells compared to non-tumor MAIT cells related to cytolysis, cytokine secretion, and metabolism of cholesterol and glucose, indicating again that both effector function and metabolic programming were affected within the HCC TME. Consistent with these findings, increased frequencies of MAIT cells within the tumor were associated with a poor prognosis [19].

The liver has a bi-directional relationship with the gut microbiota and a unique anatomical position, receiving portal blood directly from intestinal circulation and then releasing bile products back into intestines. Clinically, a strong link has been demonstrated in advanced liver disease between the gut microbiota composition and degree of hepatic encephalopathy and liver cirrhosis [87]. It is possible that in advanced cancers, the microbiome is interacting directly with the damaged hepatic tissue to pathogenically activate and exhaust MAIT cells, but this theory requires further investigation.

### 3.3. Blood Cancers

As B cells express some of the highest levels of cell surface MR1, multiple myeloma (MM) has garnered significant interest regarding MAIT cells [50]. MM patients have decreased levels of circulating MAIT cells [12,20] as well as decreased levels in the bone marrow compared to healthy controls [12,20] (Table 1). In addition, MAIT cells from patients newly diagnosed with MM had a decreased capacity to produce IFN-γ, similar to observations in other cancers. MAIT cells from Langerhans cell histiocytosis (LCH) tumors were also diminished in affected patient circulation in addition to decreased MAIT cells within the lesions [21], although MAIT cells appeared functionality unaltered (Table 1). One likely reason for the observed systemic decrease in MAIT cells is the generalized lymphocytopenia that accompanies blood cancers like MM [88]. It could also be that cancerous cells are replacing MAIT cells in the blood, or MAIT cell precursors are dying. Finally, there is evidence showing that MAIT cells themselves can become cancerous in peripheral T cell lymphomas (PTCLs). Two out of 26 cases of PCTLs were positive for PLZF staining, which were later defined as MAIT cells through identification of the TRAV1-2-TRAJ33 TCR-α rearrangement [89]. Although this cancer is rare, our increased knowledge of MAIT cell identification markers, function, activation-inducers, and inhibitors like 6-FP is paramount to our ability to treat these types of tumors.

### 3.4. Other Solid Tumors

An in vitro study culturing healthy breast MAIT cells with bacterially exposed breast carcinoma cells discovered an IL-17-biased response compared to breast MAIT cells that were stimulated with PMA-ionomycin, once more supporting the idea that the TME (or bacteria present in the TME) can alter the functionality of previously healthy MAIT cells [22] (Table 2). Other in vitro studies have shown the capacity of MAIT cells to induce cell cycle arrest [9] and cytotoxicity against MM cell lines [20] or myelogenous leukemia cell lines [10] (Table 2). While these experiments show that MAIT cells are capable of directly killing tumor cells, in vitro studies come with limitations, and there have been a lack of in vivo studies surrounding this topic.

A recent study in an in vivo melanoma mouse model found that, in contrast to the in vitro and ex vivo experiments, MAIT cells are responsible for initiating and expanding lung metastases via MR1 (Table 2) [18]. Metastases were reduced in MAIT cell-deficient MR1 knockout mice, and adoptive transfer of MAIT cells into MR1 knockouts increased lung metastases to levels similar to wild type mice, suggesting an MR1-independent mechanism. Furthermore, they suggest a mechanism that acts via a MAIT cell-dependent suppression of NK cell effector function and they highlight the potential of anti-MR1 therapy. Notably, MR1 knockout mice have been shown to have alterations in other cell types, such as γ𝛿 T cells, and mice as a species have a different MAIT subset abundance profile compared to humans [90]. As such, the results should be interpreted with caution when assessing the applicability to humans. When MAIT cells were studied in melanoma patients on anti-PD-1 therapy, high MAIT cell proportions were found in those responding to PD-1 therapy as opposed to patients who were not responding [28] (Table 1). MAIT cells from responders also had higher levels MAIT cells in the melanoma lesions than non-responders, as well as increased levels of GZMB [28] (Table 1), indicating that MAIT cells from responders have higher cytotoxic functions while circulating and infiltrating the tumor site.

## 4. MAIT Cells in Cancer Therapy

### 4.1. Chemotherapy

As other immune cells and conventional T cells are affected by chemotherapies and immunotherapies, several groups have characterized the impact of these treatments on MAIT cells. A study analyzing CD8+CD161^high^CD218^high^ T cells found that these cells had the ability to rapidly efflux various chemotherapeutic drugs, suggesting these cells may be chemo-resistant [91]. Other studies found that MAIT cells are significantly more resistant to anthracycline-containing cytotoxic drugs compared to other CD8+ T cell subsets [36,92], attributing this resistance to expression of the multi-drug resistant transporter ABCB1 (also known as MDR1). High expression of this protein is a unique feature of MAIT cells [92]. Other studies have observed that after chemotherapeutic treatment with FOLFOX (comprised of drugs Folinic Acid, Fluorouracil, and Oxaliplatin), circulating MAIT cell percentages were either unchanged or significantly higher than they were prior to surgery [9,23]. As these are percentages instead of absolute values, it is unclear whether MAIT cells are expanding or other cells are decreasing in number [9,23]. These results could suggest resistance to cytotoxic chemotherapeutic drugs, but further investigation is warranted.

### 4.2. Immunotherapy

MAIT cells express many of the targets of immune checkpoint inhibitors, highlighting the potential importance of these cells in immune checkpoint therapy. MAIT cells express PD-1 in both blood and peripheral sites [93] and enhanced expression of PD-1 has been shown on CD4+ and CD8+ T cells in some cancer patients and other disease settings [94]. Increased PD-1 levels have been found on MAIT cells in the bone marrow and peripheral blood of newly diagnosed MM (NDMM) patients compared to healthy controls, and in vitro/in vivo PD-1 blockade experiments have demonstrated successful re-activation of MAIT cells and a significant reduction in mouse tumor burden (Table 2) [12].

Other studies of squamous cell carcinoma (SCC) and basal cell carcinoma (BCC) observed that tumor-residing MAIT cells upregulated expression of IFN-γ, GZM K, GZM A and GZM M following PD-1 blockade [80]. Additionally, higher MAIT cell proportions were recently found to correlate with better responses to anti-PD-1 immunotherapy, with MAIT cells from responders having a greater cytotoxic function [28]. These results suggest that the PD-1 pathway may be involved in mediating MAIT cell dysfunction in certain cancers, and that, in addition to its canonical role in antitumor immunity, anti-PD-1 therapy may restore the expression of certain cytotoxic genes.

Studies have found a significant association between immunotherapeutic responses and commensal microbiome composition, showing that responders to anti-PD-1 therapy have favorable microbiome compositions, and vice versa [95,96]. These studies suggest that microbial modulation of MAIT cells by riboflavin biosynthesis may provide the elusive mechanistic link between the gut microbiome and systemic responses to immunotherapies. In particular, several studies have identified *Bifidobacterium longum* (*B. longum*) as the key beneficial species, which interestingly is the only *Bifidobacterium* species to produce riboflavin [97]. In studies investigating the efficacy of anti-PD-L1 therapy, *Bifidobacterium* was most significantly associated with anti-tumor effects and most abundant in those who responded to anti-PD-1 therapy [98]. After administration of this bacteria to germ-free mice, this study observed improved tumor control similar to immune checkpoint blockade, and an almost complete tumor reversal when combining oral *Bifidobacterium* treatment with an immune checkpoint inhibitor [98,99]. Furthermore, low levels of B-vitamin producing bacteria have been associated with higher risks of immunotherapy-related adverse events [100] and less MAIT cell reconstitution in patients with blood disorders after a hematopoietic cell transplantation [101]. *B. longum* specifically correlated with higher MAIT cell counts in the blood and recovery of MAIT cells after the graft [101]. Ongoing human investigations of microbiome transfers will determine whether transferring microbiota specifically enriched for vitamin-B synthesis including *B. longum* improves the efficacy of immune checkpoint inhibitors via the increased frequency or activation of MAIT cells.

## 5. Future Directions

### 5.1. MR1-Directed and CAR-T Cells Therapies

Conserved, monomorphic receptors like MR1 as well as homogenous cell types like MAIT cells are attractive targets for adoptive cell therapies, seeking to selectively induce cell death of malignant cells while leaving healthy cells untouched. MAIT cells in particular are highly prevalent in human blood making for straightforward isolation, can respond in an innate-like fashion without priming, are at less risk for off-target effects due to the tight regulation of MR1 on the cell surface, and can be easily used to target mucosal tissues like the liver and lung due to their natural tropism for those tissues [102]. Thus far, studies have focused on adoptive therapy with conventional T cells that are restricted to one HLA allele [103,104]. As there is significant variation between individual’s HLA gene loci, many patients cannot benefit from these treatments because of mismatching HLA alleles. MR1 however, is highly conserved across individuals and expressed in every human tissue, presents completely different classes of molecules, and will primarily bind to MAIT cells, eliminating the need to engineer a new TCR for each patient with a different cancer type or HLA allele.

Before successful implementation of adoptive MAIT cell therapies, several potential hurdles must be addressed. First, to successfully direct MAIT cells toward tumors, a better understanding of the nature of disease-associated antigens and antigen/MR1 complexes that could be presented in MR1 is required. As MR1 can be expressed (in varying amounts) on the surface of all cells, examining which antigens are presented in healthy and diseased tissues will be important next steps. This knowledge could also help explain MAIT cell cytotoxic effects in vitro, or, considering the increased tumor growth observed in MR1-deficient mice [18], which tumor antigen might activate MAIT cells in a pro-inflammatory or tumor-promoting state. It will also be important to examine the functions of different MAIT cell subsets so that only the optimal cells are transferred.

MR1 is also an attractive target for future therapies. Non-MAIT, MR1-restricted T cells were recently shown to recognize and kill several varieties of tumor cells in an MR1-dependent manner [105]. These results further suggest the existence of tumor-derived antigens presented by MR1. However, whether such tumor-derived compounds also contain MAIT cell antigens is currently unknown [105]. Regulation of MR1 expression is not currently well described in healthy tissues or in tumor cells, and future therapies targeting MR1 will benefit from further understanding.

In addition to therapies using MAIT cells to directly target tumors via their endogenous TCR, MAIT cells may be an ideal host for chimeric antigen receptor (CAR)-T cell therapies [106]. As MAIT cells are highly capable cytotoxic cells with distinct tissue tropisms, redirecting these functions towards other antigens may provide new treatments for difficult-to-treat tumors.

### 5.2. Other Open Questions

In addition to the questions posed above, there are still many routes to take with MAIT cell research. If adoptive and/or CAR-T therapy is shown to be effective, combining adoptive MAIT cell therapies with anti-PD-1 immunotherapies may have powerful effects. Other ideas include using TCR-technology to target multiple receptors on the target cell, MR1 and other MHC-presented antigens, in order to prevent immune-escape of cancer cells [107]. MAIT cells also have yet to be fully investigated in other difficult-to-treat tumors, such as sarcoma, mesothelioma, and glioma, as well as other common cancers such as kidney and bladder cancers, leukemia, and prostate cancers. Considering that the function and prevalence of MAIT cells varies widely between different cancer types, it will be informative to fully investigate their role. Additionally, a better understanding of the associations between MAIT cells, cancer types, and prognosis could lead to the development of a new clinical biomarkers.

Other questions remain in understanding the degree of crosstalk between MAIT cells and other immune/non-immune cells and commensal microbes when activated under normal circumstances or within the TME. It is important to note that different subtypes of cells often have different functions, and that there are likely many MAIT cell subsets that have yet to be discovered or fully elucidated. Early studies of other immune cells, such as macrophages, initially showed similar discrepancies between studies until the M1 and M2 subsets of macrophages were discovered, demonstrating anti- and pro-tumor effects, respectively [108]. There is likely a similar pattern with MAIT cells, accounting for some of the variable results discussed in this review.

As shown in this review, there are considerable discrepancies regarding the role of MAIT cells in various malignancies, but it is clear is that they do have a role and warrant further investigation.

## Figures and Tables

**Figure 1 cancers-13-01502-f001:**
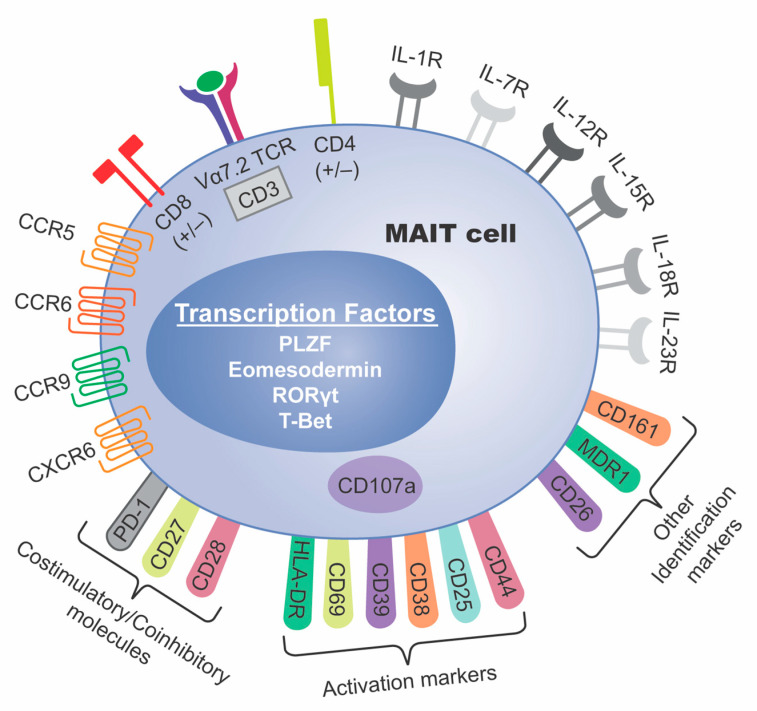
MAIT Cell Identification. A representation of known biologically significant markers found on MAIT cells. Pictured is the Vα7.2 TCR, co-stimulatory molecules CD8(+/−), CD3, CD4(+/−), CD27, and CD28, coinhibitory and immunotherapy target molecule PD-1, cytokine receptors IL-1R, IL-7R, IL-12R, IL-15R, IL-18R (CD218), and IL-23R, chemokine receptors CCR5, CCR6, CCR9, CXCR6, activation markers HLA-DR, CD69, CD39, CD38, CD25, CD44, and degranulation marker CD107a (intracellular), and other identification markers CD161, MDR1, and CD26. Identification transcription factors include PLZF, Eomesodermin, RORγt, and T-Bet.

**Figure 2 cancers-13-01502-f002:**
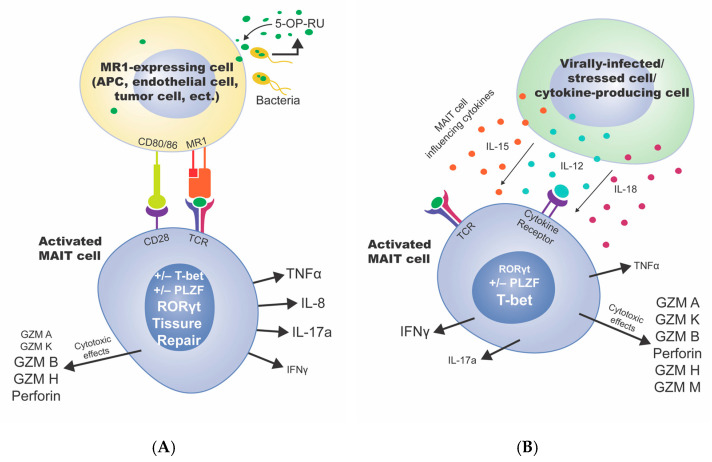
Mechanisms of MAIT cell activation. (**A**) TCR-Dependent activation via MR1 on an MR1-expressing cell (includes antigen presenting cells, endothelial cells, tumor cells, etc.). In this context, an antigen (bacterially produced 5-OP-RU) is taken up into the cell and presented in MR1. The TCR from the MAIT cell engages MR1 and other co-stimulatory molecules like CD28/CD80. Transcription factors RORγt, PLZF, T-bet, and tissue repair genes are expressed. Cytokines such as TNF-α, IL-8, IL-17a, and IFN-γ, are secreted, as well as cytotoxic effector granules like GZMA, GZMK, GZMB, GZMH, and perforin. The size of the text reflects the relative secretion of each molecule, whereas bigger text indicates more secretion and smaller text is less secretion, specific to the type of activation pictured. (**B**) TCR-Independent activation via cytokine release from other cells such as virally infected cells, stressed cells, or other cytokine-producing cells. MAIT cells are not only activated by MR1-TCR binding, but also by cytokines IL-15, IL-12, and IL-18. When the cytokine receptor is bound, cytokines TNF-α, IFN-γ, and IL-17a are secreted, as well as cytotoxic effector granules like GZMA, GZMK, GZMB, perforin, GZMH, and GZMM. Transcription factors RORγt, PLZF, and T-bet are expressed. The size of the text reflects the relative secretion of each molecule, whereas bigger text indicates more secretion and smaller text is less secretion, specific to the type of activation pictured.

**Figure 3 cancers-13-01502-f003:**
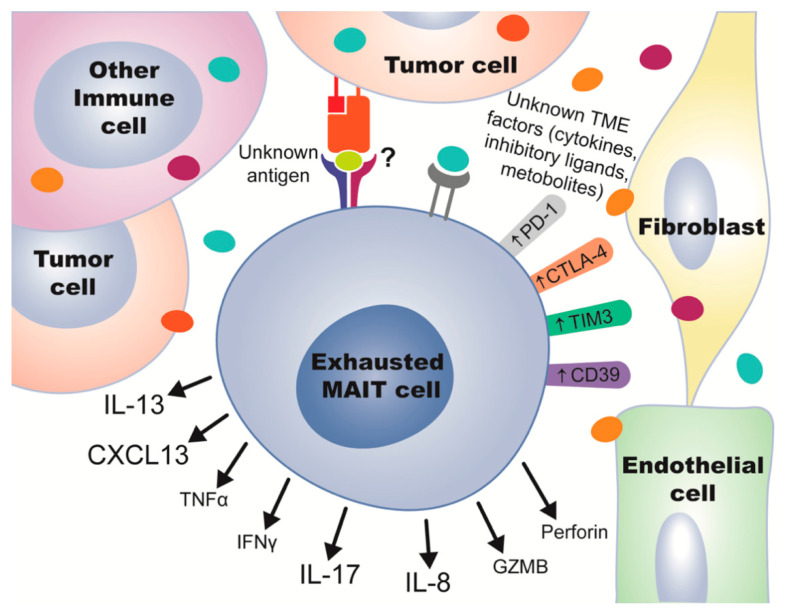
A TME-infiltrating and exhausted MAIT cell under various possible stimuli. The MAIT TCR is engaged by an unknown antigen from the TME, and other receptors are bound by unknown TME factors (possibly cytokines, inhibitory ligands, or other metabolites). Factors are potentially released by cells present in the TME such as tumor cells, fibroblasts, endothelial cells, or other immune cells. Effects from TME stimuli include upregulation of molecules PD-1, CD39, CTLA4, and TIM3, and well as release of cytokines IL-13, CXCL13, IL-17 and IL-8, with decreases in secretion of TNF-α, IFN-γ, GZMB, and perforin. The size of the text reflects the relative secretion of each molecule, whereas bigger text indicates more secretion and smaller text is less secretion.

**Table 1 cancers-13-01502-t001:** Summary of ex vivo results.

Organism Tested/Cancer Type	Method of MAIT Detection	Subjects/Tumors Assessed	MAIT Cells in Blood	MAIT Cells in Disease Site	Functionality Changes?	Association with Survival or Disease Progression	Tumor-Promoting/Tumor-Killing/Inconclusive	Reference/Year Published
Human, Brain and Kidney	Vα7.2-Jα33 TCR	8 brain, 11 kidney	MAITs detected	MAITs and MR1 detected	N/A	N/A	Inconclusive	[5]/2008
Human, CRC	CD3+, CD26+, Vα7.2+	35 CRC, 35 HAT	N/A	↑MAITs in CRC compared to HAT	N/A	↑MAITs in tumor = poor prognosis	Tumor-Promoting	[6]/2015
Human, CRC	CD45+, CD3+, CD4−, TCRγ𝛿−, CD161^high^, Vα7.2+	44 CRC, 44 HAT	No difference in MAITs between CRC and HC	↑MAITs in CRC compared to HAT (↑DN and ↓CD8+ compared to HAT)	↓IFN-γ-producing MAITs compared to HAT. No change in TNF-α or IL-17.	N/A	Inconclusive–Tumor impairs MAIT functionality	[7]/2015
Human, CRC	CD3+, TCRγδ−, CD161+, Vα7.2+	48 CRC, 22 HC	↑CD4+, ↓CD8+, ↓total MAITs compared to HC	↑MAITs in CRC compared to HC	↓IFN-γ, ↓TNF-α, ↑IL-17A in blood MAITs compared to HC	↑MAITs in early stage CRC compared to advanced CRC	Inconclusive–MAITs infiltrate tumor, tumor impairs MAIT functionality	[9]/2016
Human, MAC (colon, gastric, lung)	CD3+, TCRγδ−, CD161^high^, Vα7.2+	99 MAC, 9 HAT, 20 HC	↓MAITs compared to HC and non-mucosal cancers	↑ in CRC compared to blood and HAT	Normal cytokine profile. CCL20 and CXCL16 mRNA↑ in MAC tissue compared to HAT. ↑ corresponding CCR6 and CXCR6 on blood MAITs	↓Circulating MAITs = ↑N staging, ↑CEA	Tumor-Killing–MAITs infiltrate tumor	[10]/2016
Human, CRC	CD3ε+, CD161++, Vα7.2+, MR1-tet+	21 CRC, 21 HAT	Abundant MAITs detected	↓MAITs in CRLM compared to HAT	↓IFN-γ in CRLM MAITs compared to liver HAT	N/A	Inconclusive	[23]/2017
Human, CRC	CD45+, CD3+, TCRγδ−, CD4−, Vα7.2+, CD161^high^	35 CRC 35 HAT, 6 HC	No change between tumor and blood	↑MAITs in CRC compared to HAT	CRC MAITs had ↑ GZM B compared to blood, ↑ PFN compared to HAT. Cytotoxicity potential unchanged.	N/A	Inconclusive	[8]/2019
Human, CRC, NSCLC, RCC	Va7.2+ MR1-tet+	24 CRC, 11 NSCLC, 9 RCC	No difference between cancer types	↑MAITs in CRC compared to NSCLC and RCC. CRC had ↑ CD4+ Foxp3+ MAITs compared to HAT and blood.	CRC MAITs ↑TCR signaling and neg. apoptotic regulation pathways, and ↑CD69, CD103, CD38, and CD39 compared to blood and HAT. ↑PD-1 and CTLA-4 ON CD39+ subset.	N/A	Inconclusive–Tumor impairs MAIT functionality	[24]/2020
Human, HCC	TRAV1-2/TRAJ33, TRAV1-2/TRAJ20, or TRAV1-2/TRAJ12	6 HCC, 6 HAT	N/A	↓MAITs in HCC compared to HAT	N/A	↓MAITs = poor prognosis	Inconclusive	[11]/2017
Human, MM	CD3+, CD161+, Vα7.2+	14 NDMM, 14 HC	↓ Total, CD8+ and DN MAITs compared to HC, CD4+ MAITs unchanged	↓ Total, CD8+ and DN MAITs in BM compared to HC. CD4+ MAITs unchanged.	↓IFN-γ and ↓TNF-α compared to HC. ↑PD-1 in NDMM BM and blood compared to HC.	N/A	Inconclusive–Tumor impairs MAIT functionality	[12]/2017
Human, MM	TRAV1-2+, MR1-tet+	30 MM, 26 HC	NDMM had ↑CD4+ and DN, ↓ total and CD8+	No change in BM	NDMM MAITs had ↓IFN-γ and ↓CD27 compared to HC	N/A	Inconclusive–tumor impairs MAIT functionality	[20]/2018
Human, LCH	CD3+, Vα7.2+, CD161+, MR1-tet+	16 LCH 17 HC	↓ Total MAITs, ↑CD4+ MAITs compared to HC	↓MAITs compared to HC, MR1 detected in lesions	Normal cytokine profile	N/A	Inconclusive	[21]/2018
Human, HCC	CD3+, CD4−, CD161+, Vα7.2+, MR1-tet+	257 HCC, 257 HAT	↓CD4− MAITs compared to HC	↓CD4−MAITs compared to HAT	HCC MAITs had ↑PD-1, ↑CTLA-4, ↑TIM-3, ↑IL-18, ↓IFN-γ, ↓IL-17, ↓GZM B, ↓PFN compared to HAT and HC. ↑CCR6, ↑CXCR6, and ↑CCR9 in HCC blood compared to HCC tumor and liver	↑MAITs in tumor = poor prognosis	Tumor-Promoting	[19]/2019
Human, CC	CD3+, TCRγ𝛿−, Vα7.2+, CD161++	47 CC, 39 HC	↓MAITs in CC than in HC	N/A	N/A	Trend of ↓MAITs = more advanced stages; Trend of ↓MAITs = poor PFS	Inconclusive/Tumor-Killing	[13]/2019
Human, CC	CD3+, CD161+, Vα7.2+	36 CC, 35 controls with UL	↑IL-18, ↑CCL3/5, ↑CCR5, ↑Mo-MDSCs, ↑Tregs, ↑CD8+/↑CD4+/↑CD38+CD8+ MAITs, ↓DN MAITs, ↓DN/PD-1+ MAITs in CC	N/A	N/A	↑Total MAITs = ↑Mo-MDSCs, ↑PD-1 DN MAITs = ↑ PFS	Tumor Promoting/ Inconclusive–Depends on MAIT subset	[27]/2021
Human, EAC	CD3+, CD161^high^, Vα7.2+	79 OAC, 35 BO, 14 HC	↓ MAITs in EAC and BE compared to HC	↑ MAITs in BE and EAC compared to HAT	EAC MAITs had ↓IFN-γ, ↓TNF-α, ↓NKG2D, no change IL-17A compared to BE-adjacent tissue Tumor-conditioned HC MAITs had ↓IFN-γ and ↓TNF-α compared to MAITs in normal media	↓Tumor MAITs = ↑Risk of death	Tumor-Killing/ inconclusive–tumor impairs MAIT functionality	[14]/2019
Human, M	CD3+, CD8+, Vα7.2+, CD161+	28 M on anti-PD-1 TX	↑MAITs in responders to PD-1 TX	↑MAITs in the TME of responders compared to non-responders	MAITs from responders had ↑CXCR4 and GZMB	↑MAITs in blood = Better response to TX	Tumor-Killing	[28]/2021

BC = Breast Carcinoma, BM = Bone Marrow, BE = Barrett’s Esophagus, CC = Cervical Cancer, CEA = Carcinoembryonic Antigen, CRC = Colorectal Cancer, CRLM = Colorectal Liver Metastasis, DN = Double Negative (CD4−CD8−), GZN = Granzyme, EAC = Esophageal Adenocarcinoma, HAT = Healthy Adjacent Tissue, HBD = Healthy Breast Donors, HC = Healthy Controls, HCC = Hepatocellular Carcinoma, LCH = Langerhans Cell Histiocytosis, MAC = Mucosal Associated Cancer, M = melanoma, MM = Multiple myeloma, Mo-MDSCs = Monocytic Myeloid Derived Suppressor Cells, NDMM = Newly Diagnosed Multiple Myeloma, NSCLC = Non-Small Cell Lung Carcinoma, PFN = Perforin, PFS = Progression Free Survival, RCC = Renal Cell Carcinoma, TM = Tumor Margin, TX = Treatment, UL = Uterine Leiomyoma. The ↑ symbol indicates an increase in frequency or expression levels, and the ↓ indicates a decrease in frequency or expression levels.

**Table 2 cancers-13-01502-t002:** In vivo and in vitro results.

In Vitro/In Vivo; Cell Type	MAIT Detection	Cytokine Response	Experimental Results	Tumor Promoting/Tumor Killing/Inconclusive	Reference
In vitro; HCT116 cells: human colon carcinoma cell line	CD3+, TCRγδ−, CD161+, Vα7.2+	Activated HC MAITs produced ↑IL-17, ↑IFN-γ and ↑TNF-α when in contact with HCT116 cells	MAITs isolated from HC ↑ cell cycle arrest at G2/M phase of HCT116 cells in a dose-dependent and contact-dependent manner	Tumor-Killing	[9]
In vitro; K562 cells: myelogenous leukemia cell line	CD3+, TCRγδ−, CD161+, Vα7.2+	N/A	Activated MAITs have direct cytotoxic effect on K562 cells	Tumor-Killing	[10]
In vitro; 5T33MM: MM cell line	CD3+, CD161+, Vα7.2+	N/A	Re-activation of MAITs by PD-1 blockade → PD-1 may mediate MAIT cell dysfunction in MM. ↓Tumor load in vivo	Inconclusive/Tumor-Killing—Tumor may impair MAIT functionality	[12]
In vitro; RPMI-8226 and U266: MM cell lines	TRAV1-2+, MR1-tet+	HC MAITs had Type 1 response in killing assay → ↑IFN-γ and ↑TNF-α	5-OP-RU loaded MAITs lysed MM cells, with similar kinetics to an NK cell, in a serial killing fashion	Tumor-Killing	[20]
In vitro; MDA-MB-231: BC cell line	CD45+, CD3+, Vα7.2+, CD161+,	N/A	Bacterial-loaded BC cells ↑ IL-17 MAIT production	Inconclusive–Tumor may impair MAIT functionality	[22]
In vitro; OE33 cells: EAC cell line	CD3+, CD161^hi^, Vα7.2+	N/A	HC MAITs ↓ EAC viability	Tumor-Killing	[14]
In vivo (mouse); Melanoma: LM	MR1-tet+	N/A	↓ LM in MR1 KO, adoptive transfer of MAITs into MR1 KO ↑ LM	Tumor-Promoting	[18]

BC = Breast Carcinoma, EAC = Esophageal Adenocarcinoma, HC = Healthy Controls, KO = Knock Outs, LM = Lung Metastases, MM = Multiple Myeloma. The ↑ symbol indicates an increase in frequency or expression levels, and the ↓ indicates a decrease in frequency or expression levels.

## Data Availability

Not Applicable.

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
