# Peer review of "MAIT Cells: Partners or Enemies in Cancer Immunotherapy?"

_cancers, 2021, doi:10.3390/cancers13071502_

Round 1

Reviewer 1 Report

The authors have provided a detailed review of literature about the potential of MAIT cells and cancer therapy. Their elaborate tabulation of various aspects of the MAIT cells and their role in T cell therapy towards cancer therapy is important. Novel therapies such as CAR mediated or TCR modified therapy with MAIT cells is an interesting unexplored area. They authors could elaborate on this in the discussion of the paper.   They may have missed some references-  Lu et al., 2021 (doi: 10.1002/ijc.33411. )Zumvelde et al 2018 DOI: 10.1186/s13058-018-1036-5.,  .

Reviewer 2 Report

Unconventional T cells including MAIT cells (Mucosal-associated invariant T cells) has been significant for the better effective immunotherapy. Although its population is much lower than conventional T cells, its unique characteristics are valuable enough to investigate as a potential immunotherapy indicator against various cancers. In this review, the authors explained MAIT cells’ biology in malignant diseases. This review provided appropriate information well, but some of contents are insufficient to make readers confused, so it needs to be verified to make it clear.

 Major issues

  1. The authors should reconstitute the contents. They described ‘MAIT cell basics’ from part.1 to 5, but from ‘MAIT cell and cancer’ is not irrelevant for the explanation of MAIT cells basic information. Instead of using just ‘MAIT cell basics’ for all parts, they should add more subtitles such as ‘MAIT cell and cancer’ in part.3 and ‘Application of MAIT cells for immunotherapy’ in part.4.
  2. According to the Figure 1 in the review, only three markers (CD26, MDR1 and CD161) are described as identification markers. For readers, however, it is not sufficient to understand the standard for MAIT cell markers. Because all of the markers in the figure can be significant marker for MAIT cells. Therefore, the authors should refer other review articles about MAIT cells and add more details about MAIT cells’ biological characteristics. For example, a review article from Nature Immunology in 2019, not just CD161 and CD26 are the markers for MAIT cells, but also CD218 (IL-18 receptor) is often used for classical marker as well. It is wondering what is the standard for the markers in the given figure, and the reason why the authors described only three markers (CD161, MDR1, and CD26). In addition to Figure 2, transcription factors in each diagram should be added.
  3. The table 2 in the review is complicating to understand well. For example, expression of ‘Anti-tumor’ makes the reader confused. Reading another research papers, usually decreased level of Th1 response dose not demonstrate ‘Anti-tumor’ effect. Immune responses of MAIT cells to HCT116 cells and OE33 cells, however, the authors defined ‘Anti-tumor’ effect even the Th1 response is downregulated in the case. As IL-17 secretion, which is detrimental effect to MAIT cells, is upregulated while IFN-γ and TNF-α level is decreased. These results give advantages for tumor like immune evasion. In the case, the readers probably understand ‘Pro-tumor’ effect, but the authors described ‘Anti-tumor’ effect on the table. They need more explanation the reason why they expressed this definition, not ‘Pro-tumor’ effect. Likewise, the given table is not informative enough, they should rearrange and express much clearly.
  4. The contents of review should be rearranged. Especially, the authors seemed to introduce the interaction between commensal microbiota and MAIT cells, but it is too limited informative to make the reader figure out it. They should write it in a new chapter with more information and it would help the readers to understand well.

 Minor issues

  1. Use abbreviation after mentioning in the contents such as Nature killer cells (NK cells).
  2. Add more missing references when they mentioned the results of experiments.
  3. Checking the upper case and lower case. Ex) ‘studies’ in line 280, ‘Il-7R’ in line 140
  4. Check the comma in a sentence. Ex) line 349 in page 6
  5. Check and unify the order under the author’s guide line. Ex) reference and figure/table order

Round 2

Reviewer 2 Report

The authors have included re-written key parts of the manuscript, which is now significantly improved. They have addressed the major concerns raised in the initial review.